# Repurposing antiviral agents against mucormycosis

**Ammar A. Khan**[1,2], **Nour M. AlKashef**[1,2], **Mohamed N. Seleem**[1,2]*

1 Department of Biomedical Sciences and Pathobiology, Virginia-Maryland College of Veterinary Medicine, Virginia Polytechnic Institute and State University, Blacksburg, Virginia, United States of America, 2 Center for One Health Research, Virginia Polytechnic Institute and State University, Blacksburg, Virginia, United States of America

* seleem@vt.edu

## Abstract

Mucormycosis is a life-threatening fungal infection with limited treatment options and high mortality rates among immunocompromised individuals. To identify new therapeutic strategies, we screened a library of 618 antiviral compounds against *Rhizopus delemar* both alone and in combination with amphotericin B (AmB) to search for agents with intrinsic antifungal activity or the ability to enhance AmB's efficacy. Four candidates, IMB-301, U18666A, BLT-1, and obefazimod, showed potent in vitro effects, with three sustaining growth suppression comparable to AmB for up to 48 h in time-kill assays. The hepatitis C antivirals daclatasvir (DAC) and velpatasvir (VEL) demonstrated strong synergy with AmB across *Mucorales* isolates, lowering AmB MICs by 4- to 32-fold (ΣFICI < 0.5) and achieving fungistatic suppression of *R. delemar* at subinhibitory AmB concentrations (0.25 μg/mL). Importantly, neither the standalone antivirals nor their combinations with AmB reduced Vero cell viability at concentrations exceeding 4–16 × their MICs, while selectivity indices ranging from 8 to >32 indicated favorable safety margins. These findings highlight antiviral repurposing as a promising strategy to expand treatment options for mucormycosis and support further translational development.

## Introduction

The global incidence of invasive fungal infections has risen noticeably in recent years, driven by the expansion of at-risk populations [1,2]. Among these infections, invasive mucormycosis poses a particularly grave threat, characterized by rapid progression and high mortality. Mucormycosis is primarily caused by the inhalation of spores from the environment [3]. *Rhizopus oryzae*, recently reclassified *R. delemar*, is responsible for approximately 70% of reported cases in addition to other filamentous fungi related to the order *Mucorales* [4–6]. Individuals with compromised immune systems due to hematologic malignancies or secondary to viral infections,

**Data availability statement:** All relevant data are within the manuscript and its Supporting information files.

**Funding:** This work was supported by the National Institute of Health Grant R01AI141439. The funders had no role in study design, data collection and analysis, decision to publish, or preparation of the manuscript. The authors have declared that no competing interests exist.

**Competing interests:** None.

extensive usage of immunosuppressants, as well as other risk factors, including uncontrolled diabetes mellitus, are highly vulnerable to such invasive fungal infections [7–12]. The COVID-19 pandemic highlighted the significance of these vulnerabilities, as a global surge in mucormycosis cases was observed, particularly in the form of COVID-associated mucormycosis (CAM), which has been reported to carry mortality rates as high as 88% in some clinical settings [13,14]. In the United States, the rising prevalence of diabetes, cancer, and transplant procedures in an aging population further expands the number of individuals at risk for this often-fatal infection [15].

Amphotericin B (AmB) remains the mainstay of therapy due to its broad-spectrum antifungal activity against *Mucorales*-associated infections. However, the use of conventional AmB deoxycholate is limited by its substantial toxicity, particularly nephrotoxicity and infusion-related adverse events. While lipid-based formulations offer improved tolerability, optimal dosing strategies remain undefined, and high-dose therapy may still lead to toxicity. Alternative antifungals, including posaconazole and isavuconazole, are frequently used, yet both agents exhibit variable absorption profiles, emerging resistance, and inconsistent clinical success [5,16]. Although combination therapy is employed in severe or refractory cases, there is limited clinical evidence to support its superiority over monotherapy. Consequently, the management of mucormycosis remains challenging due to the toxicity, limited efficacy, and pharmacological constraints of available antifungals [16].

Previously, our group demonstrated the efficacy of HIV protease inhibitors against invasive fungal infections, including candidiasis and cryptococcosis, highlighting the potential of antivirals in antifungal therapy [17–23]. Since the approval of idoxuridine, the first antiviral over 50 years ago, approximately 90 antiviral drugs have been developed, many of which are FDA-approved and widely used in clinical practice [24]. Several of these antivirals possess pharmacokinetic properties that make them promising candidates for repurposing against invasive fungal infections such as mucormycosis. For instance, favipiravir demonstrates high distribution to the lungs, while acyclovir and ganciclovir exhibit effective penetration into the central nervous system, both key sites of mucormycosis [25–27].

In this study, we considered antiviral agents as potential therapeutic candidates against mucormycosis. We screened 618 approved and investigational antiviral compounds for antifungal activity against *Rhizopus delemar* strain 99–880, both as monotherapies and in combination with AmB. Compounds exhibiting standalone or adjunctive activity with AmB were further validated against a panel of *Mucorales* and non-*Mucorales* isolates and assessed for cytotoxicity in mammalian cells.

## Materials and methods

### Fungal strains & reagents

Fungal strains used in this study were obtained from the Westerdijk Fungal Biodiversity Institute, the U.S. Centers for Disease Control and Prevention (Atlanta, GA), BEI Resources (Manassas, VA), and the Cramer Laboratory at Dartmouth University. RPMI medium components included RPMI 1640 (Thermo Fisher Scientific, Waltham,

MA), 3-(N-morpholino) propane sulfonic acid (MOPS; Sigma-Aldrich, St. Louis, MO), and Potato Dextrose (PD) broth and agar (Becton, Dickinson and Company, Franklin Lakes, NJ). The Antiviral Compound Library (Catalog No. HY-L027) was purchased from MedChemExpress (Monmouth Junction, NJ). AmB was obtained from Chem-Impex International (Wood Dale, IL). Additional test compounds included IMB-301 (TargetMol Chemicals, Wellesley Hills, MA), U18666A (GlpBio, Montclair, CA), BLT-1, obefazimod, velpatasvir, and DAC (Ambeed, Arlington Heights, IL). MTS reagent was obtained from BOC Sciences (Shirley, NY) and PMS reagent was obtained from TCI Chemicals (Tokyo, Japan).

### Screening of the antiviral compound library and identification of compounds

To identify potential antifungal compounds, we screened the MedChemExpress (MCE) Antiviral Compound Library against *R. delemar* RA 99–880 under two conditions: (1) compounds tested individually at a fixed concentration of 16 µM in RPMI-1640 medium, and (2) compounds tested in combination with sub-inhibitory AmB at 0.0625 µg/mL (0.125 × MIC). Fungal inoculum was prepared from 5- to 7-day-old cultures grown on potato dextrose agar and diluted to a final concentration of ~$5.0 \times 10^4$ CFU/mL. Each well received 100 µL of inoculated medium. Plates were incubated at 35°C for 24 hours. Fungal growth was assessed by measuring optical density at 530 nm using Tecan F200 Pro spectrophotometer (Tecan Trading AG, Switzerland).

### *In vitro* MIC determination and checkerboard assay

The CLSI M38/M27 guideline was used to determine the MIC values of IMB, BLT, U18 and OBE against the fungal isolates [28,29]. The *in vitro* interactions between AmB, DAC and VEL were evaluated against all isolates, using the microdilution checkerboard assay, as previously described [18,30–32]. Both MIC and checkerboard readings were determined at 24–96 hours post-incubation, depending on the strain and species, and the wells selected displayed 100% growth inhibition. The FICI was calculated and interpreted as follows: FICI values ≤ 0.5 were categorized as synergism (SYN), those between 0.5 and 4 as indifference (IND), and those >4 as antagonism (ANT).

### Time kill assay

The effect of the positive hits from antiviral library screening on the growth kinetics of *R. delemar* was performed, as previously described for filamentous fungi [30]. Fungal spores were diluted in RPMI-MOPS medium to a final concentration of $5 \times 10^4$ conidia/mL. The cultures were then treated with BLT-1, IMB-301, U18666A, VEL, and DAC at 1× and 2 × MIC. AmB was included as a positive control, and untreated wells containing only fungal spores served as negative controls. Fungal growth was monitored by measuring optical density at 530 nm ($OD_{530}$) using a Tecan F200 Pro Multi Mode Plate Reader at 0, 6, 12, 24, and 48 hours. Each condition was tested in five independent wells.

### Cytotoxicity assessment of test compounds and combinations in vero cells

The cytotoxic effects of the test compounds, IMB-301, BLT-1, U18666A, obefazimod, were evaluated individually using Vero CCL-81 cells (African green monkey kidney epithelial cells), while DAL and VEL were additionally assessed in combination with AmB to evaluate potential additive toxicity [18,33]. Cells were seeded into flat-bottom 96-well microtiter plates at a density of $2 \times 10^5$ cells/well in Eagle's Minimum Essential Medium (EMEM) supplemented with 10% fetal bovine serum (FBS) and 1% penicillin-streptomycin. After a 24-hour incubation at 37 °C in a humidified atmosphere with 5% $CO_2$ to allow cell adherence, the culture medium was replaced with fresh EMEM containing serial dilutions of the test compounds, either alone or in fixed-ratio combinations with AmB (1 µg/mL). Control wells received an equivalent volume of DMSO. Following 24 hours of drug exposure, cell viability was assessed using the MTS/PMS assay by incubating cells with the reagent mix for 3 hours. The absorbance of the resulting formazan product was measured at 490 nm using a Tecan F200 Pro Multi Mode Plate Reader. All treatments were performed in biological triplicates. Selectivity indices (SI) were calculated as the ratio of the 50% cytotoxic concentration ($CC_{50}$) to the minimum inhibitory concentration (MIC) for

 

each compound (SI = CC$_{50}$/MIC) [34]. CC$_{50}$ values were derived from Vero cell cytotoxicity assays and are reported as approximate values based on observed ranges.

## Statistical analysis

All statistical analyses were conducted using GraphPad Prism version 8.0 (La Jolla, CA, USA). Statistical analyses were performed using one-way ANOVA followed by Dunnett's multiple comparison test to evaluate differences among treatment groups. Significance was defined as p ≤ 0.05.

## Results

### Screening of antivirals for individual and potentiating effects with AmB on *Rhizopus delemar*

To identify compounds with potent inhibitory activity against *Rhizopus delemar* 99−880, we screened the MedChem Express (MCE) Antiviral Compound Library, comprising 618 clinical and investigational antiviral agents. Compounds were tested in RPMI-MOPS medium, buffered at pH 7, at a fixed concentration of 16 µM alone and in the presence of a sub-inhibitory concentration of AmB (0.0625 µg/mL). Hits were identified based on either (i) ≥70% inhibition of fungal growth as determined by quantitative optical density measurements or (ii) visual inspection of assay wells showing complete or near-complete fungal clearance, even if they did not meet the 70% OD inhibition threshold. Using these combined criteria, 17 compounds were identified with individual anti-fungal activity against the tested strain (Fig 1A). Based on prior literature, 13 compounds were excluded due to known cytotoxicity or previously reported antifungal activity (S1 Table). The remaining four hits with standalone activity, IMB-301, U18666A, BLT-1, and obefazimod, were selected for further

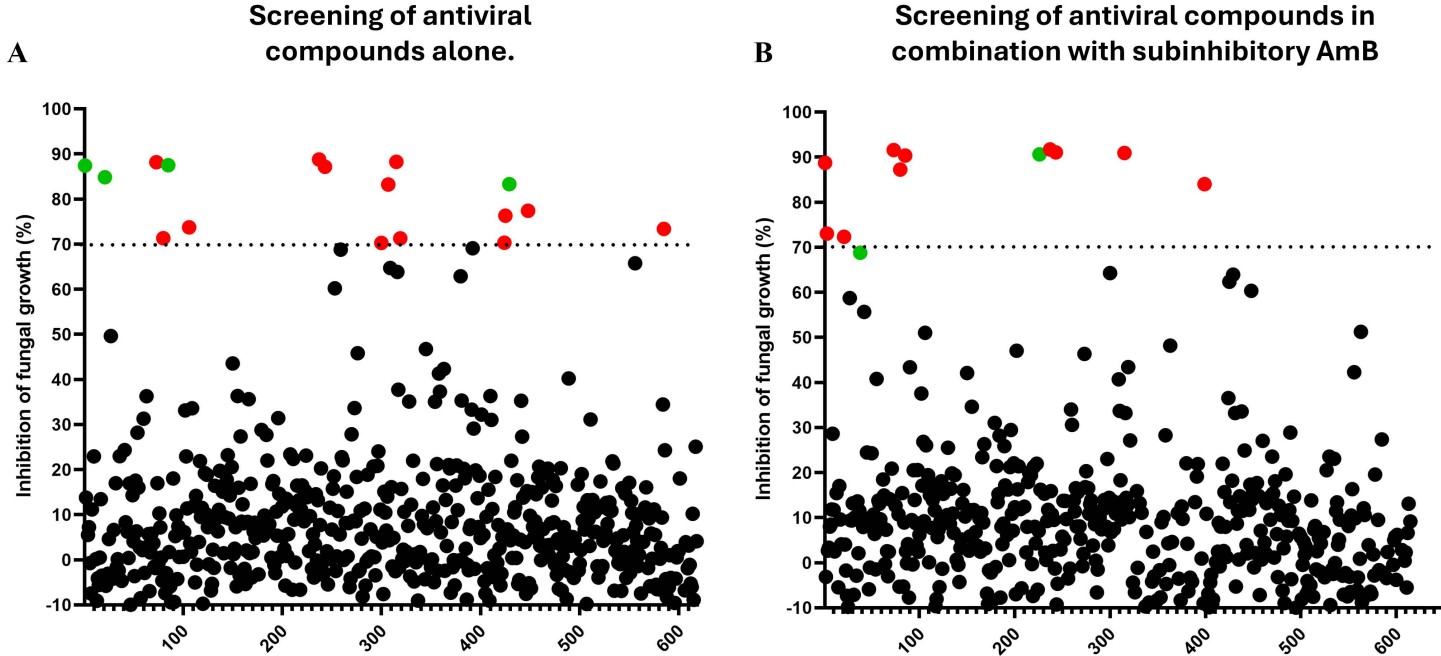

**Fig 1. Identification of antivirals compounds with potential antifungal activity against *Mucorales* causing fungi *R. delemar*.** (A) Screening of the MedChemExpress (MCE) Antiviral Compound Library at a final concentration of 16 µM against *R. delemar* 99-880. (B) Screening of the same library in the presence of a subinhibitory concentration of amphotericin B (0.0625 µg/mL). Fungal growth was assessed after 24 hours by measuring the optical density at 530 nm. Red and green dots represent the selected antivirals and other excluded compounds eliciting ≥70% growth inhibition relative to the control growth.

validation (Fig 1A) and are shown with their chemical structures and brief functional descriptions (Fig 2). As part of our screening to identify antiviral compounds that enhance AmB activity, the hepatitis C antivirals daclatasvir (DAC) and velpatasvir (VEL) were identified as promising potentiators. DAC achieved approximately 91% inhibition of *Rhizopus delemar* growth compared to AmB treatment alone, while VEL demonstrated 69% inhibition by optical density but met the criteria for a hit based on marked visual suppression of fungal growth (Fig 1B). Notably, several compounds that initially appeared as inhibitors in the standalone screen did not retain activity in the AmB-supplemented screen. This change suggests that certain combinations may interfere with AmB's activity, warranting further investigation into potential antagonistic interactions.

**Evaluation of potent hits against *Mucorales* and non-*Mucorales* clinical isolates using the broth microdilution**

The antifungal activity of the four most potent hits, IMB-301, BLT-1, U18666A, and obefazimod, was further evaluated against a panel of 14 fungal strains. Among the *Mucorales* isolates (n = 8), all four compounds exhibited $MIC_{50}$ values of 4 µg/mL, though obefazimod showed greater variability across strains (Table 1). tested fungal panel included several non-*Mucorales* pathogens, including *Scedosporium* and *Lomentospora* species, filamentous fungi known for intrinsic resistance to AmB and poor clinical outcomes [35]. Notably, BLT-1 demonstrated modest activity against select strains of *Scedosporium* and *Lomentospora*, suggesting possible strain-specific effects outside the *Mucorales* order. In contrast, IMB-301 and U18666A showed limited or no activity against *Candida auris*, *Aspergillus fumigatus*, and *Cryptococcus neoformans*. These findings suggest that the antifungal effects of the identified compounds are primarily directed against *Mucorales*, with BLT-1 exhibiting the broadest spectrum within the group.

**BLT-1**
Scavenger Receptor B, Type 1 inhibitor

**U18666A**
Cholesterol transport inhibitor

**IMB-301**
HIV-1 replication inhibitor

**Obefazimod**
HIV-1 replication inhibitor

**Daclatasvir**
HCV NS5A inhibitor

**Velpatasvir**
HCV NS5A inhibitor

**Fig 2. Chemical structures of lead antiviral hits identified from the primary screen.** Standalone hits (IMB-301, BLT-1, U18666A, and obefazimod) demonstrated intrinsic antifungal activity against Rhizopus delemar 99-880, while daclatasvir and velpatasvir exhibited synergistic interactions with amphotericin B.

**Table 1. Description of the clinical isolates used in the study and in vitro activity of antiviral hits.**

| Strain | Origin | Source | MIC (µg/mL) | | | | |
|---|---|---|---|---|---|---|---|
| | | | BLT-1 | IMB-301 | U18666A | Obefazimod | AmB |
| *Rhizopus delemar* RA 99–880 | USA | Patient with rhinocerebral mucormycosis [36] | 4 | 4 | 2 | 8 | 1 |
| *Rhizopus arrhizus var. delemar* CBS 120593 | France | Human Lung Isolate [37] | 4 | 4 | 2 | 8 | 1 |
| *Mucor indicus* CBS 123974 | Germany | Patient with gastric mucormycosis [37] | 4 | 4 | 4 | 4 | 1 |
| *Mucor janssenii* CBS 243.67 | South Africa | Patient with mucormycosis [37] | 8 | 8 | >16 | 4 | 0.5 |
| *Mucor circinelloides* 1006PhL | USA | Skin sample from normal human volunteer [38] | 8 | 8 | >16 | 4 | 1 |
| *Lichtheimia ornata* CBS 142195 | China | Human nasal soft tissue [37] | 8 | 8 | 8 | >32 | 1 |
| *Lichtheimia hyalospora* CBS 146577 | Netherlands | Human skin biopsy sample [37] | 4 | 4 | 4 | 4 | 1 |
| *Rhizomucor miehei* CBS 147454 | Netherlands | Human lung tissue from an oncology patient [37] | 4 | 4 | 4 | 2 | 0.5 |
| *Lomentospora prolificans* CBS 120312 | Belgium | Lung biopsy from patient with cystic fibrosis [37] | 8 | 16 | >16 | >32 | 8 |
| *Lomentospora prolificans* CBS 742.96 | Netherlands | Sample from patient with leukemia [37] | 4 | 16 | 16 | >32 | 8 |
| *Scedosporium aurantiacum* (CBS 136046) | Australia | Sample from patient with lung disease [37] | 4 | 0.25 | 16 | >32 | 4 |
| *Scedosporium apiospermum* CBS 129968 | China | Human lung isolate [37] | 2 | 16 | 16 | >32 | 4 |
| *Cryptococcus neoformans* H99 ATCC208821 | USA | Sample from patient with Hodgkins disease [39] | 4 | 0.125 | 2 | >32 | 0.5 |
| *Candida auris* AR0390 | Japan | Ear discharge from human patient [40] | 8 | 16 | 16 | >32 | 2 |
| *Aspergillus fumigatus* AF293 | UK | Lung biopsy [41] | >16 | >16 | >16 | >32 | 2 |

## Evaluation of velpatasvir and daclatasvir synergy with AmB via checkerboard microdilution assay

We assessed the in vitro interactions between AmB and the antivirals VEL and DAC against a panel of twelve clinically relevant *Mucorales* and non-*Mucorales* strains (Table 2). Both AmB/VEL and AmB/DAC combinations demonstrated consistent synergy across all *Mucorales* isolates, with fractional inhibitory concentration index (ΣFICI) values for AmB/VEL ranging from 0.04688 to 0.14063, and for AmB/DAC from 0.09375 to 0.26563, indicating substantial enhancement of AmB activity.

Among AmB–resistant *Lomentospora* and *Scedosporium* strains, most combinations were indifferent (FICI~0.5–1); however, notable synergistic interaction was observed between AmB and VEL against *S. apiospermum* and with AmB/DAC combination against *L. prolificans* 120312. Furthermore, both combinations showed synergistic activity against additional high-priority fungal pathogens, including *Candida auris*, *Cryptococcus neoformans*, and *Aspergillus fumigatus*, with ΣFICI values ranging from 0.265 to 0.3125 (Fig 3), highlighting their broader therapeutic potential.

## Effect of antivirals on growth kinetics of *R. delemar* 99–880

To evaluate the effects of selected antiviral compounds on the growth of *Rhizopus delemar*, a time-kill assay was conducted using conidial suspensions ($5 \times 10^4$ conidia/mL) of strain *R. delemar* 99–880. Cultures were treated with BLT-1, IMB-301, U18666A, and obefazimod at 2 × MIC concentrations, alongside AmB (1 µg/mL) as a positive control. The tested antiviral compounds showed modest or delayed inhibitory effects over the 48-hour period. Remarkably, BLT-1, IMB-301, and U18666A suppressed fungal growth to a similar extent as AmB throughout the entire assay duration (Fig 4A and 4B; S2 Data). In contrast, obefazimod exhibited temporary growth suppression, maintaining inhibition for the first 24 hours

**Table 2. Checkerboard microdilution of amphotericin B (AmB) interaction in combination with velpatasvir (VEL) and daclatasvir (DAC) against various *Mucorales* isolates.**

| Strain | Data for AmB/VEL | | | | Data for AmB/DAC | | | |
|---|---|---|---|---|---|---|---|---|
| | MIC (µg/mL) | | ΣFICI | Mode | MIC (µg/mL) | | ΣFICI | Mode |
| | Alone | Combined | | | Alone | Combined | | |
| *Rhizopus delemar* (RA 99–880) | 1/>128 | 0.125/4 | 0.15625 | SYN | 1/128 | 0.125/8 | 0.1875 | SYN |
| *Rhizopus arrhizus var. delemar* (CBS 120593) | 1/>128 | 0.125/2 | 0.14063 | SYN | 1/>128 | 0.125/4 | 0.15625 | SYN |
| *Mucor indicus* (CBS 123974) | 1 >128 | 0.0625/2 | 0.07813 | SYN | 1/>128 | 0.25/2 | 0.26563 | SYN |
| *Mucor janssenii* (CBS 243.67) | 0.5/>128 | 0.03125/2 | 0.07813 | SYN | 0.5/>128 | 0.0625/2 | 0.14063 | SYN |
| *Mucor circinelloides* (1006PhL) | 1/>128 | 0.03125/2 | 0.04688 | SYN | 1/>128 | 0.125/2 | 0.14063 | SYN |
| *Lichtheimia ornata* (CBS 142195) | 1/>128 | 0.125/2 | 0.14063 | SYN | 1/>128 | 0.125/4 | 0.15625 | SYN |
| *Lichtheimia hyalospora* (CBS 146577) | 1/>128 | 0.03125/2 | 0.04688 | SYN | 1/>128 | 0.125/2 | 0.14063 | SYN |
| *Rhizomucor miehei* (CBS 147454) | 0.5/>128 | 0.03125/2 | 0.07813 | SYN | 0.5/>128 | 0.0625/2 | 0.14063 | SYN |
| *Lomentospora prolificans* (CBS 742.96) | 8/>128 | 8/2 | 1.0156 | IND | 8/>128 | 8/4 | 1.031 | IND |
| *Lomentospora prolificans* (CBS 120312) | 8/>128 | 4/2 | 0.5156 | IND | 8/>128 | 2/8 | 0.3125 | SYN |
| *Scedosporium aurantiacum* (CBS 136046) | 4/>128 | 2/8 | 0.5625 | IND | 4/>128 | 2/8 | 0.5625 | IND |
| *Scedosporium apiospermum* (CBS 129968) | 4/>128 | 1/2 | 0.2656 | SYN | 4/>128 | 2/8 | 0.5625 | IND |

The fractional inhibitory concentration index (FICI) was calculated and interpreted as follows: FICI values ≤ 0.5 were categorized as synergism (SYN), those between 0.5 and 4 as indifference (IND), and those >4 as antagonism (ANT)

before fungal growth resumed. These findings indicate varying degrees of fungistatic activity among the tested compounds, with the first three showing sustained inhibition comparable to AmB.

Similarly, the effect of AmB/VEL and AmB/DAC on the growth of *R. delemar* was assessed using conidial suspensions in RPMI-MOPS medium. Cultures were treated with AmB at a subinhibitory concentration (0.25 µg/mL) either alone or in combination with DAC or VEL (8 µg/mL). Both combinations markedly suppressed *R. delemar* growth for up to 24 hours (Fig 4C and 4D), achieving inhibition comparable to that of AmB alone at its MIC$_{90}$ (1 µg/mL). However, growth resumed thereafter, indicating that these combinations exerted a strong but transient fungistatic effect.

### Cytotoxicity effects of standalone and combination antivirals on mammalian cells

Cytotoxicity was assessed for each antiviral compound at concentrations exceeding their antifungal MICs using kidney epithelial cells Vero CCL4. AmB, tested at 8 µg/mL significantly reduced cell viability compared to untreated controls, consistent with its narrow therapeutic window. In contrast, standalone antivirals tested at 32 µg/mL showed no significant change in viability relative to controls (Fig 5A). Likewise, DAC and VEL, tested at 8 µg/mL individually and in combination with AmB (1 µg/mL), did not significantly affect cell viability (Fig 5B). Selectivity indices indicated favorable safety margins: U18666A (SI > 32), BLT-1 (SI > 32), IMB-301 (SI ≈ 16), and obefazimod (SI ≈ 8). These results confirm that the tested antivirals maintain safety at concentrations above their MICs and that AmB combinations with DAC or VEL do not increase cytotoxicity.

### Discussion

Mucormycosis remains one of the most challenging invasive fungal infections to manage clinically. These infections are marked by aggressive angioinvasion, leading to vascular thrombosis and widespread tissue necrosis. This pathology facilitates rapid dissemination to distant organs and is associated with alarmingly high mortality rates, ranging from 50% to nearly 100%, particularly among immunocompromised patients [42,43].

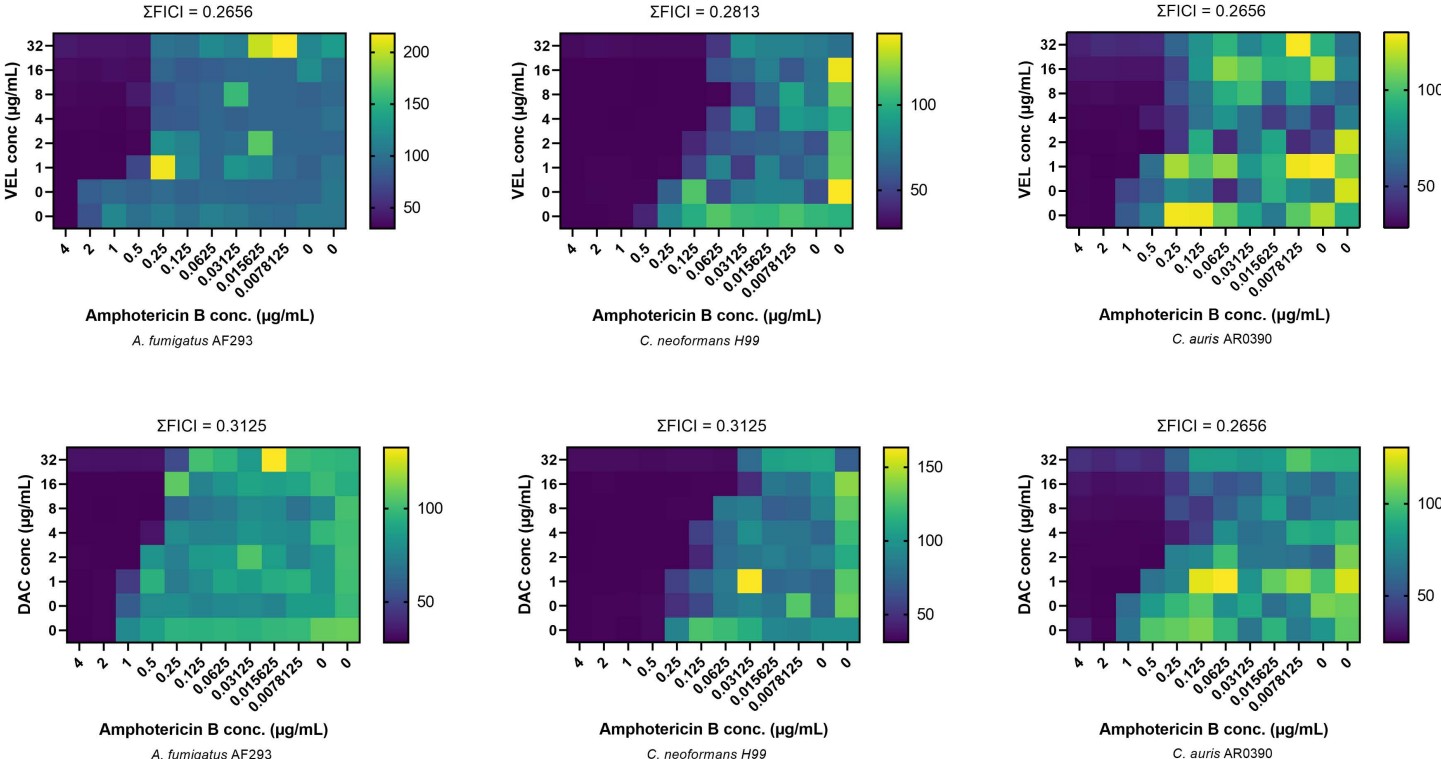

**Fig 3. Synergistic activity of VEL and DAC antivirals in combination with AmB against *A. fumigatus* AF293, *C. neoformans* H99, and *C. auris* AR0390.** The first row shows the interaction between AmB and VEL, while the second row shows the interaction between AmB and DAC. Color intensity corresponds to the fungal growth relative to the untreated cells, as measured spectrophotometrically after 24 hours.

The global incidence of mucormycosis has been rising steadily, driven by an expanding population of at-risk individuals, including those with uncontrolled diabetes, immune suppression, and prolonged corticosteroid use [44]. This trend was dramatically amplified during the COVID-19 pandemic, particularly during the second wave in early 2021. A striking surge in mucormycosis cases, especially in India, was observed among patients with current or prior SARS-CoV-2 infection, a phenomenon now referred to as COVID-19–associated mucormycosis (CAM) [43,45,46].

Compared to other invasive fungal infections, therapeutic options for mucormycosis are more limited. Currently, only two antifungal classes, polyenes and azoles, are routinely employed in clinical practice, yet these treatment options remain limited and are frequently associated with significant toxicity, suboptimal efficacy, and high failure rates, underscoring the urgent need for safer and more effective therapies [16].

Given the slow pace of antifungal drug development and the urgent need for more effective treatment options, drug repurposing offers a pragmatic and accelerated path forward [47].

In this study, we adopted a systematic drug repurposing strategy by screening a library of 618 antiviral compounds against *R. delemar* 99–880. Antivirals represent an attractive class for repositioning due to their extensive clinical use, favorable pharmacokinetics, and known safety profiles. Many antivirals demonstrate excellent oral bioavailability, tissue distribution to mucormycosis-relevant sites such as lungs and CNS, and a reduced risk of drug-drug interactions relative to some antifungals [24–27,48].

From our primary screen, we identified four compounds with potent standalone antifungal activity: IMB-301, U18666A, BLT-1, and obefazimod. IMB-301 is a small molecule that inhibits HIV-1 replication by blocking the virus's ability to

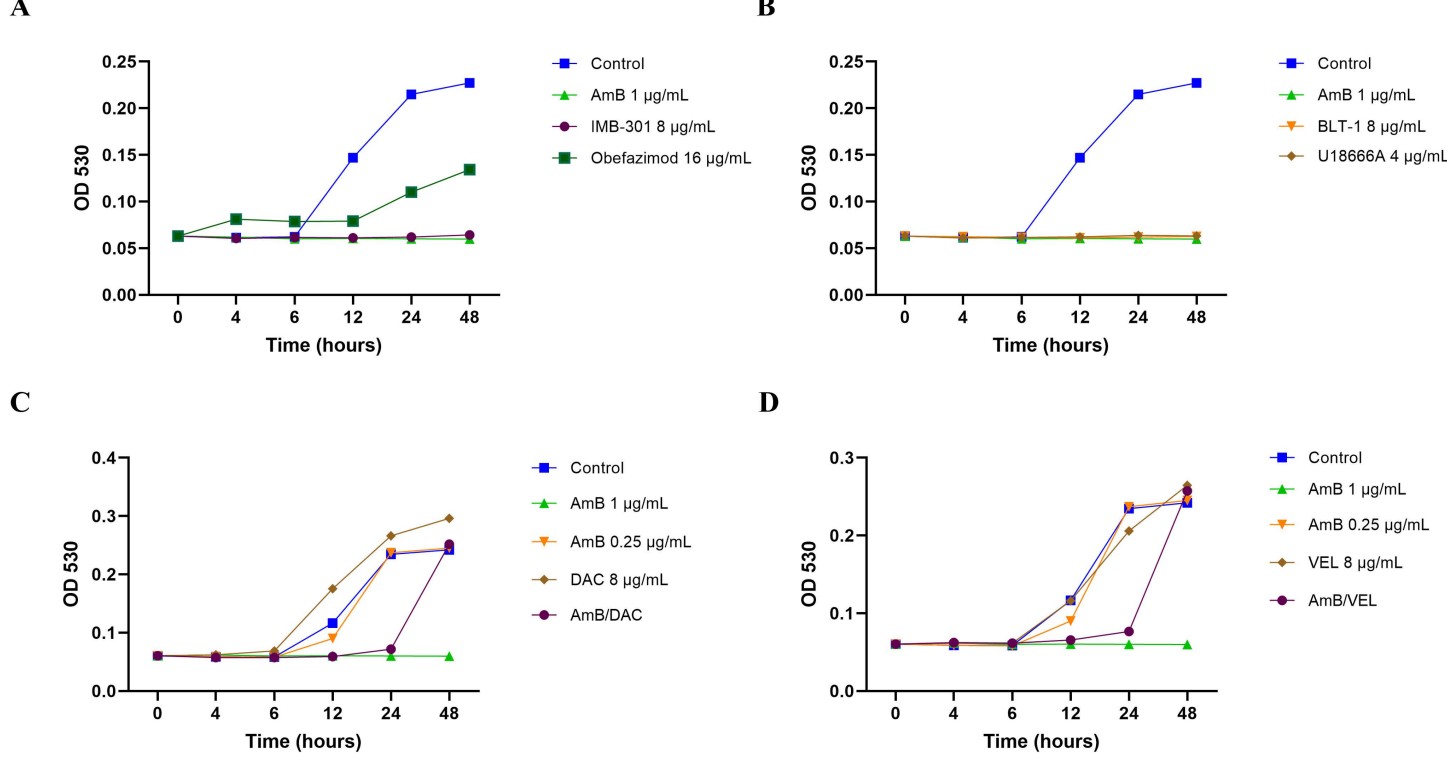

**Fig 4. Effect of antivirals on the growth kinetics of *Rhizopus delemar* 99−880.** (A and B) Proliferation of conidial suspension of *R. delemar* 99−880 (5 x $10^4$ conidia/mL) in RPMI-MOPS medium in the presence of IMB-301, BLT-1, U18666A, and obefazimod (2x MIC). (C and D) the potentiating activity of VAL and DAL (8 μg/mL) combined with sublethal concentration of AmB (0.25 μg/mL). AmB (1 μg/mL) was involved as a positive control. Optical density of treated cultures was measured at 530 nm at different time points in a 48-hour period.

degrade a key host antiviral protein. It shows low cytotoxicity in human T-cell lines at effective concentrations ($IC_{50} \sim 8.6$ μM) but has not been evaluated in vivo [49].

U18666A exhibits broad-spectrum pathogen inhibitory activity by disrupting cholesterol homeostasis. It inhibits replication and entry of viruses such as Dengue and Ebola by interfering with cholesterol-dependent trafficking pathways [50,51]. It also protects host cells from *Clostridioides difficile* toxins by blocking the uptake of cholesterol-dependent toxins [52]. Notably, U18666A accumulates in lysosomes and crosses the blood-brain barrier, allowing for CNS exposure, a property beneficial for targeting invasive fungal pathogens, such as *Mucorales*, which frequently cause rhino-cerebral infections [53,54].

BLT-1 is a highly potent SR-BI inhibitor with nanomolar efficacy, proven to block cholesterol uptake and disrupt HCV entry at low concentrations [55,56]. It also exhibits copper-chelating activity, which could be useful given the essential role of copper in fungal virulence and survival, including in *Mucorales* [57,58].

Obefazimod was initially developed as a novel antiviral therapy for HIV-1, designed to overcome the limitations of conventional antiretrovirals such as resistance and inability to eliminate viral reservoirs [59]. Since its development, obefazimod has been evaluated more broadly in inflammatory and infectious diseases, including ulcerative colitis and COVID-19. In these clinical trials, it was administered orally at daily doses ranging from 50 mg to 150 mg and demonstrated a favorable safety profile [60,61]. Although it did not show additional benefit in COVID-19, it was well tolerated in over 500 patients [62]. Obefazimod also increases levels of the anti-inflammatory microRNA miR-124 and accumulates in gastrointestinal tissues such as rectal mucosa [60]. Given that gastrointestinal mucormycosis commonly affects the stomach and intestines, obefazimod's accumulation in these tissues may have clinical relevance [63].

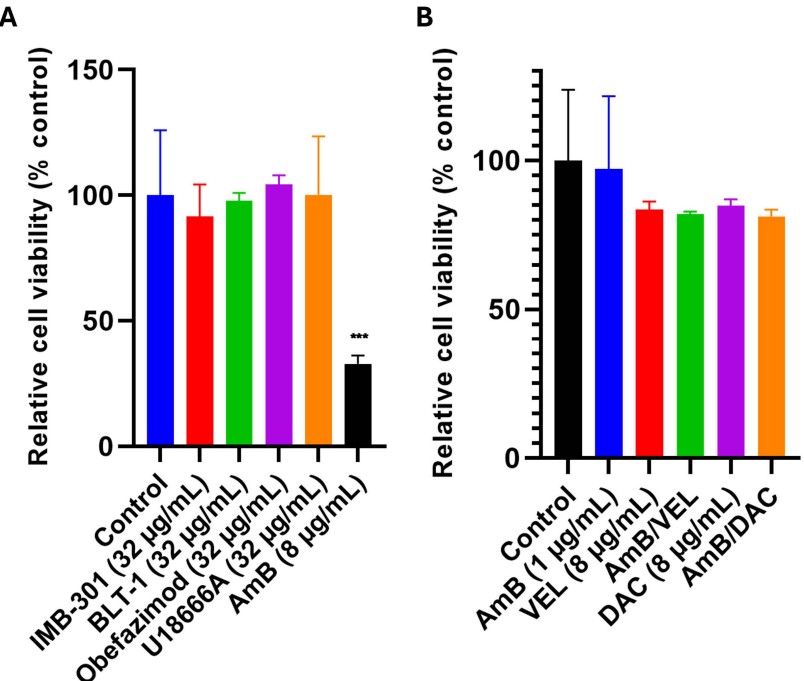

**Fig 5. Cytotoxicity of antiviral compounds on Vero kidney epithelial cells.** (A) Cytotoxicity of IMB-301, BLT-1, U18666A, and obefazimod at 32 µg/mL (B) Cytotoxicity of daclatasvir and velpatasvir (8 µg/mL) alone and in combination with AmB (1 µg/mL. In both panels, Vero cells were exposed for 24 hours, and viability was assessed using the MTS/PMS colorimetric assay, with absorbance measured at 490 nm. Results are expressed relative to untreated controls. Asterisks indicate a statistical significance ($P$ value < 0.05, as determined by one-way analysis of variance [ANOVA] using Dunnett's test for multiple comparisons) compared to the untreated control.

Among the standalone antivirals identified, IMB-301 and U18666A demonstrated the most consistent and broad-spectrum activity against *Mucorales* isolates, each with a $MIC_{90}$ value of 8 µg/mL and 4 µg/mL, respectively. BLT-1 also exhibited notable activity, with a $MIC_{90}$ of 8 µg/mL, effectively inhibiting several *Mucor* species. Obefazimod showed greater variability across isolates, with a $MIC_{90}$ exceeding 16 µg/mL; however, it retained appreciable potency against select strains, including *Mucor janssenii* and *Mucor circinelloides*.

These findings were further supported by time-kill assays, in which IMB-301, U18666A, and BLT-1 achieved sustained suppression of *Rhizopus delemar* growth over 72 hours, closely paralleling the antifungal efficacy of AmB. In contrast, obefazimod induced an early fungistatic effect that waned after 24 hours, indicating transient inhibition.

In addition to standalone antifungal hits, we identified two HCV antivirals, velpatasvir (VEL) and daclatasvir (DAC), that significantly potentiated the activity of AmB in vitro. These NS5A inhibitors are characterized by high oral bioavailability and favorable safety profiles [48]. VEL was developed as a broad-spectrum HCV agent and has also been repurposed for SARS-CoV-2 treatment. In clinical trials, it is administered orally at a daily dose of 100 mg (often with sofosbuvir) and has demonstrated excellent tolerability, including in patients with hepatic impairment [64–66]. Similarly, DAC is administered at 60 mg daily, also in combination with sofosbuvir, and has shown potent antiviral activity across multiple HCV genotypes along with good tolerability in individuals with advanced liver disease [67–69]. It too has been investigated for repurposing against SARS-CoV-2 [69].

The observed synergy between AmB and these two antivirals highlights a promising combination strategy for mucormy-cosis. Checkerboard microdilution assays showed that both AmB/VEL and AmB/DAC combinations consistently enhanced AmB activity across all tested *Mucorales* strains, reducing AmB MICs by 8- to 32-fold. This effect was further supported

by time-kill assays, where both combinations, even with AmB at subinhibitory concentrations (0.25 µg/mL), suppressed *Rhizopus delemar* growth to a level comparable with that of full-dose AmB monotherapy till 24 hours.

Importantly, the antifungal activity of these agents was not accompanied by increased cytotoxicity at concentrations exceeding their antifungal MICs. AmB, tested at clinically relevant levels, significantly reduced Vero cell viability, consistent with its well-known nephrotoxic liability. In contrast, all standalone antivirals maintained cell viability even at multiples of their MIC values, and combinations of DAC or VEL with AmB did not exacerbate AmB-associated cytotoxicity. Selectivity index estimates further supported these findings, with U18666A and BLT-1 displaying the most favorable safety margins, followed by IMB-301 and obefazimod. These data suggest that the identified antivirals not only enhance or contribute to antifungal efficacy but also exhibit safety characteristics that could mitigate one of the major limitations of current mucormycosis therapy.

This study has some limitations. First, all antifungal and synergistic activities were evaluated in vitro, and these results may not fully reflect the pharmacological behavior or therapeutic potential of these compounds in vivo. Although the identified antivirals demonstrated robust inhibition of *Mucorales* and strong potentiation of AmB, additional studies in animal models will be necessary to determine whether these findings translate to meaningful improvements in disease outcome. Second, mechanistic studies were not performed and will be important for defining how these agents exert antifungal effects or enhance AmB activity.

Overall, our findings highlight the promise of antiviral drug repurposing as a strategic avenue for addressing the urgent therapeutic gaps in mucormycosis treatment. The identified compounds, whether acting alone or in synergy with AmB, demonstrated antifungal activity and could offer valuable starting points for further investigation. These agents may serve as valuable starting points for medicinal chemistry optimization or for uncovering novel anti-fungal mechanisms, an approach that could ultimately lead to safer, more effective therapies for a disease that remains exceptionally difficult to treat.

## Supporting information

**S1 Table. Description of hits from screening of antiviral compound library.**
(DOCX)

**S2 Data. In-vitro activity of antiviral compounds.**
(XLSX)

## Acknowledgments

We acknowledge the CDC and BEI Resources for providing the fungal isolates used in this study.

## Author contributions

**Conceptualization:** Ammar A. Khan, Mohamed N. Seleem.

**Data curation:** Ammar A. Khan.

**Formal analysis:** Ammar A. Khan.

**Funding acquisition:** Mohamed N. Seleem.

**Investigation:** Ammar A. Khan.

**Methodology:** Ammar A. Khan.

**Project administration:** Mohamed N. Seleem.

**Supervision:** Mohamed N. Seleem.

**Validation:** Ammar A. Khan.

**Visualization:** Ammar A. Khan.

**Writing – original draft:** Ammar A. Khan, Nour M. AlKashef.

**Writing – review & editing:** Ammar A. Khan, Nour M. AlKashef, Mohamed N. Seleem.

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
