## [Decision Letter · Decision Letter 0]

5 Sep 2025

Dear Dr. Seleem,

Thank you for submitting your manuscript to PLOS ONE. After careful consideration, we feel that it has merit but does not fully meet PLOS ONE’s publication criteria as it currently stands. Therefore, we invite you to submit a revised version of the manuscript that addresses the points raised during the review process.

**ACADEMIC EDITOR: Please, verify the Reviewers´comments to improve the quality of the manuscript. One Reviewer suggested a reference, please, carefully verify if it is or not necessary in the manuscript.**

We look forward to receiving your revised manuscript.

Kind regards,

Otávio Augusto Chaves

Academic Editor

PLOS ONE

Journal Requirements:

“This work was supported by the National Institute of Health Grant R01AI141439.”

“None”

4. Please ensure that you refer to Figure 2 in your text as, if accepted, production will need this reference to link the reader to the figure.

6. If the reviewer comments include a recommendation to cite specific previously published works, please review and evaluate these publications to determine whether they are relevant and should be cited. There is no requirement to cite these works unless the editor has indicated otherwise.;

Reviewers' comments:

Reviewer's Responses to Questions

**Comments to the Author**

1. Is the manuscript technically sound, and do the data support the conclusions?

Reviewer #1: Yes

Reviewer #2: Yes

2. Has the statistical analysis been performed appropriately and rigorously?

Reviewer #1: I Don't Know

Reviewer #2: N/A

3. Have the authors made all data underlying the findings in their manuscript fully available?

Reviewer #1: Yes

Reviewer #2: Yes

4. Is the manuscript presented in an intelligible fashion and written in standard English?

Reviewer #1: Yes

Reviewer #2: No

Reviewer #1: 1. Please upload Institutional Ethics Committee clearance report.

2. How many times were each experiment repeated ?

3. Screening for antifungal activity in Figure 1 was done to a final time of 24 hours while time kill assays were read up to 48 hours. Why this discrepancy.

4. Please label Figure 1 A and Figure 1 B better to make them self explanatory.

5. Heading of Figure 1 refers to the fungi as Mucor - it should be Mucorales - also note were these experiments not conducted on any nnon-Mucorales like Scedosporium

6. Was any hormetic effect ever detected in any of the time kill assays?

7. Mention limitations of your study at the end of Discussion.

Reviewer #2: The manuscript entitled "Repurposing Antiviral Agents Against Mucormycosis" by Khan et al. represents a follow up study of several previous studies published under references no. [17-23] by the same group. They highlight the efficacy of HIV protease inhibitors against invasive fungal infections, including candidiasis and cryptococcosis, and the potential of antivirals in antifungal therapy. The authors screened a library of a total of 618 antiviral compounds against

Rhizopus delemar in the presence of a subinhibitory concentration of amphotericin B (0.0625 μg/mL) and performed microdilution to determine the minimal inhibitory concentrations for antifungal activity of the four most potent hits, IMB-301, BLT-1, U18666A, and obefazimod against a panel of 14 fungal strains (Mucorales, n=8) by microdilution. Cytotoxicity was measured on kidney cells. The compounds selected appeared to be highly effective against Rhizopus in in vitro antifungal activity testing, whilst cytotoxicity is low. However, information on the mode of action is missing.

Mucorales like Rhizopus are known to evade the immune system. Spores could survive within professional phagocytes like macrophages and monocytes and escape from antifungal accessibility.

How does administration of these compounds alter the intracellular fungal clearance of Rhizopus in professional phagocytes. A killing assay displaying the intracellular accessibility of these antiviral drugs within macrophages is highly recommended.

Ref. for immune evasion:

Morales-Franco B, Nava-Villalba M, Medina-Guerrero EO, Sánchez-Nuño YA, Davila-Villa P, Anaya-Ambriz EJ, Charles-Niño CL. Host-Pathogen Molecular Factors Contribute to the Pathogenesis of Rhizopus spp. in Diabetes Mellitus. Curr Trop Med Rep. 2021;8(1):6-17. doi: 10.1007/s40475-020-00222-1. Epub 2021 Jan 22. PMID: 33500877; PMCID: PMC7819772.

**Do you want your identity to be public for this peer review?** For information about this choice, including consent withdrawal, please see our Privacy Policy

Reviewer #1: **Yes:** Shiv Sekhar Chatterjee

Reviewer #2: No

---

## [Author Response · Author response to Decision Letter 1]

18 Dec 2025

The "Response to Reviewer" letter is part of the new upload package.

---

## [Decision Letter · Decision Letter 1]

4 Jan 2026

Dear Dr. Seleem,

Thank you for submitting your manuscript to PLOS ONE. After careful consideration, we feel that it has merit but does not fully meet PLOS ONE’s publication criteria as it currently stands. Therefore, we invite you to submit a revised version of the manuscript that addresses the points raised during the review process.

We look forward to receiving your revised manuscript.

Kind regards,

Otávio Augusto Chaves

Academic Editor

PLOS One

Journal Requirements:

If the reviewer comments include a recommendation to cite specific previously published works, please review and evaluate these publications to determine whether they are relevant and should be cited. There is no requirement to cite these works unless the editor has indicated otherwise.;

Additional Editor Comments:

Please, provide the synergy plot that indicate the positive synergism in the proposed combination of drugs. The authors might use SynergyFinder 2.0 software as a recommendation.

Reviewers' comments:

Reviewer's Responses to Questions

**Comments to the Author**

Reviewer #1: All comments have been addressed

2. Is the manuscript technically sound, and do the data support the conclusions?

Reviewer #1: Yes

3. Has the statistical analysis been performed appropriately and rigorously?

Reviewer #1: Yes

4. Have the authors made all data underlying the findings in their manuscript fully available?

Reviewer #1: Yes

5. Is the manuscript presented in an intelligible fashion and written in standard English?

Reviewer #1: Yes

Reviewer #1: 1. Kindly explain why different subinhbitory concentration of Amphotericin B were used during time kill assays (0.25 microgram / microliter) and screening of agent (0.0625 microgram/microliter) synergy tests

**Do you want your identity to be public for this peer review?** For information about this choice, including consent withdrawal, please see our Privacy Policy

Reviewer #1: **Yes:** Shiv Sekhar Chatterjee

---

## [Author Response · Author response to Decision Letter 2]

20 Jan 2026

Reviewer comments have been addressed in the the file titled "Response to Reviewers 2". Cover letter with amended statement also attached.

---

## [Editor Report · Decision Letter 2]

26 Jan 2026

Repurposing Antiviral Agents Against Mucormycosis

PONE-D-25-46540R2

Dear Dr. Seleem,

We’re pleased to inform you that your manuscript has been judged scientifically suitable for publication and will be formally accepted for publication once it meets all outstanding technical requirements.

Kind regards,

Otávio Augusto Chaves

Academic Editor

PLOS One
---

## [Editor Report · Acceptance letter]

PONE-D-25-46540R2

PLOS One

Dear Dr. Seleem,

I'm pleased to inform you that your manuscript has been deemed suitable for publication in PLOS One. Congratulations! Your manuscript is now being handed over to our production team.

Kind regards,

on behalf of

Dr. Otávio Augusto Chaves

Academic Editor

PLOS One